# Study of Aerogel-Modified Recycled Polyurethane Nanocomposites

**DOI:** 10.3390/nano13182583

**Published:** 2023-09-18

**Authors:** Xiaohua Gu, Shangwen Zhu, Siwen Liu, Yan Liu

**Affiliations:** 1School of Energy and Building Environment, Guilin University of Aerospace Technology, Guilin 541004, China; glzsw2337899@163.com; 2State Key Laboratory for Modification of Chemical Fibers and Polymer Materials, College of Materials Science and Engineering, Donghua University, Shanghai 200051, China; 3College of Innovative Material and Energy, Hubei University, Wuhan 430062, China; 202121113012770@stu.hubu.edu.cn

**Keywords:** waste polyurethane foam, SiO_2_ aerogel, alcoholysis, two-component decrosslinker, chemical degradation

## Abstract

In this study, a liquid regenerated polyether polyol was obtained after the degradation of waste PU foam by the two-component decrosslinker agents ethylene glycol and ethanolamine. The regenerated polyol-based polyurethane foam was modified by adding different ratios of SiO_2_ aerogel through the self-preparation of silica aerogel (SiO_2_ aerogel) to prepare aerogel/regenerated polyurethane foam nanocomposites of SiO_2_ aerogel-modified regenerated polyurethane composites. A series of analytical tests on self-prepared silica aerogel and aerogel-modified recycled polyurethane foam composites were performed. The analysis of the test results shows that the regenerated rigid PU foam obtained with SiO_2_ aerogel addition of 0.3% in the polyurethane degradation material has a small density, low thermal conductivity, and higher compressive strength; hence, the prepared silica aerogel-regenerated polyol-based polyurethane nanocomposite has good thermal insulation and strength support properties. The clean, low-carbon, and high-value utilization of recycled waste polyurethane was achieved.

## 1. Introduction

Polyurethane (PU) is a block copolymer composed of alternating hard and soft segments synthesized by the reaction between the O-H (hydroxyl) group of the polyol and the N=C=O (isocyanate functional group) group of isocyanate, which belongs to the family of thermoplastic elastomers [1,2]. Due to its excellent performance, polyurethane is being produced on a global scale and is known as the “fifth major plastic”, becoming an important part of polymer materials. Among polyurethane consumption, polyurethane foam (PUR) is equivalent to 67% of global polyurethane consumption [3]. Moreover, due to the very mature production technology, this type of foam corresponds to half of the overall polymer foam market [4]. The variable chemical composition of polyurethane foam can change the surface properties and thus adjust the interaction between the foam and other substances [5], which makes polyurethane foam enter multiple fields of production and life nowadays [6], such as insulation material for the outer layer of houses and decorative material for the interior of houses [7,8], insulation compartments for refrigerators [9], and insulation layers for pipes [10].

The sustainability of polyurethane plastics involves two main aspects. The first one is the replacement of non-renewable components with renewable energy sources. This main objective consists of limiting the extraction and processing of fossil raw materials and reducing the environmental degradation caused by this process. The second objective is to reduce non-degradable waste emissions to the environment [11]. In 2019, the International Union of Pure and Applied Chemistry (IUPAC) published a list of the top 10 emerging technologies in chemistry that have the potential to make our planet more sustainable. Among them, technologies that allow the conversion of plastic materials into monomers are highlighted, and these recycling technologies will help reduce plastic waste and save fossil resources [12]. The polyurethane foam market was valued at $46.8 billion in 2014 and $72.2 billion in 2020, and the development of polyurethanes from renewable and environmentally friendly raw materials has become a growing topic of research due to sustainability and other environmental concerns [13].

Because of the widespread use of polyurethane foam, it is produced in large quantities, and at the same time, a large amount of waste foam is generated in the process of industrial production and use [14,15]. There are three main types of disposal technologies for polyurethane foam waste: landfill [16], incineration [17], and recycling [18]. Incineration and landfill produce a large number of pollutants [19], which harm the natural environment and seriously affect people’s normal life. Recycling is divided into physical recycling and chemical recycling [20,21]. Chemical recovery methods can be divided into hydrolysis, acidolysis, ammonolysis, and alcohololysis. Alcoholysis with polyfunctional alcohols gives a mixture of products with two phases, polyether polyol in the upper phase and more polar alcoholic solvents and aromatic compounds in the lower phase; the polyether polyol recovered from the upper phase is used to synthesize new foams [22,23,24].

Meanwhile, with the increasing concern about global energy consumption, new solutions to reduce energy loss are attracting more and more attention, and silica aerogel has entered the picture as a material capable of reducing heat loss. Silica aerogel is a nano-porous solid material formed by a three-dimensional network of air-based, amorphous silica particles with a complex three-dimensional network structure, which includes properties such as low thermal conductivity, low density, high porosity, high light transmission, high specific surface area, low refractive index, and low sound velocity, and thus has a wide range of applications in thermal insulation, catalyst carriers, energy conservation, sewage purification, biomedical engineering, and aerospace exploration [25,26,27].

Polyurethane foam production is highly dependent on petroleum, and the degraded material obtained by degrading polyurethane foam can replace part of the raw material. To improve the chemical recycling of waste polyurethane, one must obtain high-quality recycled polyurethane foam and improve the performance of the foam, often adding some fillers to it, so that the prepared composite foam has a broader application prospect in building insulation [28]. In terms of thermal conductivity, low-density closed-cell polyurethane foams are the best traditional insulation materials used in construction, and they are mainly used in floor, wall, and roof insulation panels, but also as spray insulation [29]. Insulation is an effective method to reduce building heat loss, obtain the ideal temperature in rooms, and improve the quality of the indoor environment and the health of residents [30]. Aerogel is a synthetic porous material that replaces the liquid component of the gel with a gas while maintaining the gel structure [31]. Aerogel material, as an emerging nanoscale material, is widely used in various aspects of life for its unique porous and low thermal conductivity properties [32,33,34]. To enhance the thermal insulation and other properties of polyurethane foam, aerogel materials can be applied to the foaming process of polyurethane to prepare high-value recycled polyurethane foam materials.

Polyurethanes often form composites with synthetic fibers (glass, aramid, and carbon) and natural fibers, which can improve their mechanical properties [35]; also, polyurethanes can be combined with inorganic materials in forming composites. For example, in our previous work we combined mullite and polyurethane to form mullite/polyurethane composites, which improved the overall performance of the material [36].

In this study, silica aerogel was prepared at room temperature and pressure using ethyl orthosilicate, anhydrous ethanol, and dimethylformamide as raw materials and added to the preparation of regenerated polyurethane foam to produce silica aerogel-regenerated polyol-based polyurethane foam nanocomposites. We degraded waste polyurethane material and reused the degraded material, which is unprecedented in the field of polyurethane recycling. This experiment generated regenerated polyurethane foam nanocomposites with better performance than traditional polyurethane foam while reducing polyurethane foam waste, reducing the production cost, and increasing the product’s value, which is of great significance to achieving carbon emission reduction and promoting energy saving.

## 2. Materials and Methods

### 2.1. Materials and Reagents

The materials used mainly include ethyl orthosilicate (Shanghai Aladdin Biochemical Technology Co., Ltd., AR, Shanghai, China), anhydrous ethanol (Tianjin Comio Chemical Reagent Co., Ltd., AR, Tianjin, China), hydrochloric acid (Sinopharm Group Chemical Reagent Co., Ltd., AR, Shanghai, China), ammonia (Nanjing Chemical Reagent Co., Ltd., AR, Nanjing, China), dimethylformamide (Tianjin Comio Chemical Reagent Co., Ltd., AR, Tianjin, China), ethylene glycol (Jiangsu Yongfeng Chemical Reagent Co., Ltd., AR, Jiangsu, China), ethanolamine (Jinan Mingliang Chemical Co., Ltd., AR, Jinan, China), polyether 4110 (Shandong Vanke New Materials Co., Ltd., AR, Shandong, China), triethanolamine (Nanjing Renheng Chemical Co., Ltd., AR, Nanjing, China), dibutyltin dilaurate (Xindian Chemical Materials Co., Ltd., AR, Kaohsiung, China), dimethyl silicone oil (Jinan Silicon Port Chemical Co., Ltd., AR, Jinan, China), cyclopentane (Shandong Xuchen Chemical Technology Co., Ltd., AR, Shandong, China), and isophorone diisocyanate (IPDI) (Shandong Lyon New Material Technology Co., Ltd., AR, Shandong, China).

### 2.2. Preparation of Silicon Dioxide Aerogel

The silica aerogel samples were prepared quickly and conveniently at room temperature and pressure using ethyl orthosilicate (TEOS) as the silica source and anhydrous ethanol (EtOH) as the solvent. Ethyl orthosilicate and anhydrous ethanol were added to the beaker and stirred well using a magnetic stirrer. A certain amount of distilled water was added, and dilute hydrochloric acid was added dropwise to adjust the pH to the range of 3–5; then, the solution was further stirred for 30 min, and the stirred solution was placed in a thermostatic water bath with a preset temperature of 40 °C and left to stand for more than 20 min. The solution after acid hydrolysis was placed on a constant temperature magnetic stirrer, to which dimethylformamide was added dropwise and then stirred for 30 min; then distilled water was added and stirred for 30 min, and the pH was adjusted to between 8 and 10 by adding diluted ammonia dropwise to the resulting solution and stirring for 30 min to make it fully react. The stirred solution was poured into a beaker and left to stand for 2 h at room temperature, at which time the silica alcohol gel was obtained.

The beaker containing silica alcohol gel was sealed with cling film and aged; then, water and anhydrous ethanol were added to the already aged beaker for aging treatment; then, it was rinsed with hexane, the treated silica alcohol gel was dried, and the white solid obtained was silica aerogel. The mechanism and experimental flow are shown in Figure 1 and Figure 2. The dosage of drugs for this experiment is shown in Table 1.

### 2.3. Principles of Preparation of Silicon Dioxide Aerogel-Regenerated Polyol-Based Polyurethane Nanocomposites

The waste PU foam was degraded by using ethylene glycol and ethanolamine as the raw materials and then recycled. The waste PU foam was cleaned and dried and then crushed and weighed at 100 g, and 100 g of alcohololytic agent and 1 g of catalyst KOH were added and placed in a three-neck flask to react at 180 °C for a certain period of time; the degradation products obtained were subjected to certain treatments to obtain recycled polyether polyol. In order to improve the performance of oligomeric polyol foam such as heat insulation, the recycled polyurethane foam was modified by adding a certain amount of silica aerogel, and the specific proportions are shown in Table 2. The properties of the recycled polyurethane foam formulation were further improved by adding different proportions of silica aerogel, and the silica aerogel-regenerated polyol-based polyurethane foam nanocomposite was prepared.

Based on the results of the previous part of the experiments, performance enhancement experiments were conducted using the prepared regenerated polyol, polyether polyol 4110, dimethylsilicone oil, cyclopentane, triethanolamine, dibutyltin laurate, IPDI, and different amounts of silica aerogel as raw materials for the previous part of the experiments.

Totals of 30.00 g of polyol, 0.60 g of dimethylsilicone oil, 0.30 g of triethanolamine, and 0.15 g of dibutyltin dilaurate were added, after which different masses of silica aerogel were added in disposable plastic cups; the plastic cups were then placed on an ultrasonic shaker and shaken and mixed for 1 h, after which 9 g of cyclopentane was added to the mixture. We then used a cantilever mixer at a stirring rate of 600 r/min and added 39 g of polyisocyanate to the polyol mixture and stirred until foaming. The regenerated polyether polyol-based silica aerogel-regenerated polyurethane composite foam was placed in a cool place for 72 h, and various tests were performed. The specific formulations are shown in Table 2.

Since silica aerogel contains some silicon hydroxyl groups on its surface when the hydroxyl group of the polyol reacts with the isocyanate group in the IPDI, some of the silicon hydroxyl groups of silica aerogel will replace the hydroxyl group of the polyol and combine with the isocyanate group to form the carbamate group, which becomes the skeleton part of polyurethane foam. The silica aerogel that does not contain the hydroxyl groups is incorporated into the regenerated polyurethane foam and becomes a physical reinforcing agent. The reaction mechanism is shown in Figure 3.

## 3. Results and Discussion

### 3.1. Fourier Transform Infrared Spectroscopy

The component silica aerogel specimens were tested and analyzed via infrared spectroscopy using Spectrum-one (American PE Co., Ltd, Denver, CO, USA) and the result is shown in Figure 4. The sample preparation methods are listed in Table 1. It can be seen from the figure that the vibration peak of the Si-O-Si group in the silica aerogel molecule appears near 1089 cm^−1^, and this peak is more acute, and there are obvious peaks at 464 cm^−1^ and 800–850 cm^−1^, which are the bending vibration peaks of the Si-O-Si group and symmetric stretching vibration peaks [37,38], indicating that the products contain a large amount of the Si-O-Si bond; the absorption peaks appearing near 3400 cm^−1^ are -OH group vibration peaks, indicating the presence of a certain amount of -OH groups in the specimens. In summary, the main components of the prepared specimens were all silica aerogels.

As shown in Figure 4, there is a strong absorption band at 1510 cm^−1^ for the methylene (-CH_2_) absorption band between the two benzene rings, a strong absorption band at 1224 cm^−1^ for the ester group (C=O) absorption band of aromatic polyurethane, and an absorption band at 1084 cm^−1^ for the C-O-C group of polyether type polyurethane, so it can be determined that the polyol used in the preparation of waste polyurethane foam is a polyether type polyol and the isocyanate is diphenylmethane diisocyanate (MDI) or its prepolymer.

### 3.2. XRD Test Analysis

The prepared silica aerogels were tested and analyzed using an X-ray diffraction analyzer (Empyrean, Malvern Panaco Co., Ltd, Amsterdam, The Netherlands). The test radiation tube had a voltage of 40 kV, a current of 180 mA, a scanning range of 5–90°, and a speed of 3.5°/min. The silica aerogel is mainly composed of silica, and it can be seen from Figure 5 that both silica aerogel sample 1 and silica aerogel sample 2 have the same diffuse diffraction peaks, indicating that both the self-prepared silica aerogel and commercial silica aerogel are disordered amorphous bodies dominated by silica [39].

### 3.3. BET Analysis

The specific surface adsorption–desorption isothermal curves of the aerogel samples were determined on a BET automated adsorbent apparatus (Micromeritic Tristar 3000) at a temperature of 77 K. The adsorbent was N_2_. The test results are shown in Figure 6.

After testing, the BET-specific surface area of the aerogel sample was 624.95 m^2^·g^−1^. Figure 6 shows the adsorption–desorption isothermal curves of N_2_ adsorption by the aerogel sample. The results show that the interaction between the adsorbent and the aerogel sample is very weak and the adsorption of N_2_ on the surface of the aerogel is a multimolecular layer adsorption at the beginning, and the curve shows an asymptote parallel to the longitudinal axis when the pressure is close to saturated vapor pressure P_0_, which is a typical manifestation of the coalescence of the adsorbent of N_2_ between solid particles. The result is consistent with the structural characteristics of the aerogel, which is a porous solid coalesced by silica nanoparticles. This result is consistent with the structural characteristics of the aerogel, which is a porous solid condensed by silica nanoparticles, and the adsorption isotherm and desorption isotherm are in good agreement with each other, which indicates that the pore size of the aerogel is large, and the distribution of the pore size is narrow.

### 3.4. Effect of Different Reactants on the Density of Silica Aerogel

Using a vibration density meter to measure the density of silica aerogel powder, a certain amount of specimen powder was put into the measuring cylinder, the cylinder was fixed on the vibration device, it was vibrated for 12 min, and we obtained the result that the volume of the specimen after the vibration no longer changed. According to the formula *ρ* = *m*/*V*, one can obtain the density of silica aerogel powder. Figure 7 shows that the vibration density of the silica aerogels prepared using different ethyl orthosilicate and anhydrous ethanol concentrations is slightly different. Therefore, the best ratio consists of 30 g of ethyl orthosilicate, 45 g of anhydrous ethanol, and 5 g of distilled water, which has the smallest vibration density of 0.0043 g/cm^3^.

### 3.5. Scanning Electron Microscope Analysis

The produced silica aerogel was characterized using a scanning electron microscope (SU660, Hitachi, Tokyo, Japan) to observe the morphology of the produced silica aerogel. The test characterization is shown in Figure 8. It can be seen that the microstructure of the silica aerogel shows a three-dimensional network of the skeleton, and these silica particles stack on each other to form the skeleton structure. When the mass ratio of ethyl orthosilicate, anhydrous ethanol, and distilled water was 15:45:5, the mass ratio of water was slightly larger, which made some of the already formed silica–oxygen bonds re-hydrolyze, so that the silica particles of the prepared silica aerogel were not uniform enough, as shown in (a). When the mass ratio of ethyl orthosilicate, anhydrous ethanol, and distilled water was 30:45:5, it can be seen from (b) that the silica particles were closely stacked and the skeleton was more complete; additionally, some white agglomerates appeared when the mass of ethyl orthosilicate gradually increased, as shown in (c) and (d), which is due to the high ratio of ethyl orthosilicate and the polymerization rate being too fast, which makes the silica atoms agglomerate, and at this time, due to the agglomeration of silica particles, the larger the proportion of ethyl orthosilicate, the greater the possibility of breaking the three-dimensional mesh structure of silica aerogel products, so that the distribution of silica particles is no longer uniform, and the performance of the silica aerogel will be reduced.

### 3.6. Infrared Spectroscopy of Silicon Dioxide Aerogel-Regenerated Polyol-Based Polyurethane Nanocomposites

The IR spectra of the regenerated polyol-based silica aerogel-regenerated polyurethane nanocomposites prepared with commercial polyether 4110 are shown in Figure 9. It can be seen from Figure 9a that the spectra of the regenerated polyurethane nanocomposites with different silica aerogel additions are approximately the same as those of the unadded regenerated polyurethane foam, with some differences only at some vibrational peaks. From the figure, it can be seen that the stretching vibrational peaks appear as hydroxyl and N-H near 3325 cm^−1^. At 2960 cm^−1^, there is a stretching vibrational peak of C-H in methyl or methylene [40], and in the degradation products, a small amount of amino groups form salts due to the cleavage of amino ester bonds, resulting in the appearance of NH^+^ and NH^2+^ peaks at 2255 cm^−1^. The peak at 1745 cm^−1^ is the vibrational peak of the C=O bond in the carbamate group [41], the peak at 1513 cm^−1^ is the stretching vibrational peak of -NH, there is a more pronounced peak at 1240 cm^−1^ for the stretching vibrational peak of C-N, and the stretching vibrational peak of the ether bond is near 1050 cm^−1^ [26]. In summary, it can be seen that the IR spectra of the regenerated polyurethane foam prepared using regenerated polyol and adding silica aerogel are the same as those of the polyurethane nanocomposites prepared from polyether 4110, indicating that the specimens prepared using regenerated polyol are polyurethane foams.

From Figure 9b, it can be seen that there is a symmetric stretching vibration peak of the Si-O-Si bond at 766–815 cm^−1^, a fine antisymmetric stretching vibration peak of the Si-O-Si bond at 1065 cm^−1^, and an absorption peak at about 517 cm^−1^ due to the bending vibration of the Si-O-Si bond [25,42]. The shift of the peak near 1090 cm^−1^ in the foam with the addition of silica aerogel is caused by the combination of the antisymmetric absorption peak of Si-O-Si at 1065 cm^−1^ in its vicinity. In summary, it is shown that the addition of silica aerogel affects the preparation of polyurethane nanocomposites.

### 3.7. Elemental Analysis of Silicon Dioxide Aerogel-Regenerated Polyol-Based Polyurethane Nanocomposites

The test results of the X-ray photoelectron spectroscopy analysis are shown in Figure 10. Figure 10a shows the XPS spectra of the 4110 polyurethane foam and the regenerated polyurethane foam, from which it can be seen that the elemental compositions are very similar with only minor differences. The peaks at 285.8 eV (C1s), 399.8 eV (N1s), and 532.3 eV (Ols) are distinctive peaks in the XPS spectra of the 4110 polyurethane foam. Figure 10b shows the XPS spectra of silica aerogel-regenerated polyurethane nanocomposite with 0.3% of the total mass of polyol and regenerated polyurethane foam without silica aerogel addition. The silica aerogel-regenerated polyurethane nanocomposite not only contains characteristic peaks at the abovementioned positions but also has Si 2s and Si 2p peaks. The corresponding C1s, N1s, and O1s peaks are also relatively enhanced, indicating that the silica aerogel has become part of the urethane group and can enhance the performance of the regenerated polyurethane nanocomposites.

### 3.8. Compression Strength Test Analysis

The compressive strength of the specimen was tested using a universal material testing machine, and the silica aerogel-regenerated polyurethane composite foam with different silica aerogel additions was made into a 50 × 50 × 50 mm specimen block and measured at a displacement speed of 20 mm/min; multiple identical samples were repeated, and the average value was taken. The test results are shown in Figure 11.

It can be seen from Figure 11 that the compressive strength of the silica aerogel-regenerated polyol-based polyurethane foam nanocomposites changed with the addition of silica aerogel. Since the aerogel has a highly permeable cylindrical multi-branched nano-porous triple network structure, it can increase the support strength of the bubble pores in the regenerated polyurethane foam and enhance the compressive strength to a certain extent; when the silica aerogel is added in excess, the agglomeration phenomenon will occur and make the bubble pore structure of the foam uneven, and then the support performance of the foam pore wall will be reduced and the compressive strength will be lowered. Therefore, adding the right amount of silica aerogel to the recycled polyurethane foam will increase the compressive strength. From the test, it can be learned that when the addition amount is 0.09 g (0.3% of the total polyol), the compression strength of regenerated polyether polyol-based silica aerogel-regenerated polyurethane nanocomposite is 0.585 MPa, which reaches the maximum value, at which time the compression strength best reaches the best performance. Additionally, when the amount of silica aerogel addition continues to increase, leading to excess, the silica aerogel addition of 0.12 g and 0.15 g will incur an agglomeration phenomenon, so that the support properties of the pore wall of the foam decreases, leading to a reduction in strength.

### 3.9. Apparent Density Test Analysis

The specimens of silica aerogel-regenerated polyurethane nanocomposites with different additions of silica aerogel were left in a cool place (25 °C) for 72 h, and then the regenerated polyurethane specimens were cut into 50 × 50 × 50 mm specimen blocks to test their apparent density, and the specimen blocks were weighed using an electronic balance to calculate the apparent density of the specimens according to the formula *ρ* = *m*/*V*. The experiment was repeated five times for each group of formulations and the average value was taken; the measurement results are shown in Figure 12.

As can be seen from Figure 12, the lowest density of the regenerated polyurethane foam specimen with the addition of silica aerogel, 32.1 kg/m^3^, is similar to that of 34.1 kg/m^3^ without the addition of silica aerogel, with only a slight change. As the three-dimensional mesh structure of silica aerogel becomes part of the polyurethane skeleton, the pore size of the bubble pores will increase, resulting in a slight increase in the density of the regenerated polyurethane foam. With the increase in the amount of silica aerogel, some silica aerogel particles may form agglomerates, which makes the regenerated polyurethane nanocomposites fluctuate more; the density of regenerated polyurethane nanocomposites fluctuates least when the silica aerogel addition is 0.09 g, and the density is 32.8 kg/m^3^ at this time.

### 3.10. Thermal Conductivity Test Analysis

The thermal conductivity was measured using a thermal conductivity tester (DSC-725L, Shanghai Jiubin Instrument Co., Ltd., Shanghai, China). Polyurethane foam is commonly used as insulation material in daily life, so thermal conductivity is an important index for polyurethane foam. Due to the low thermal conductivity of aerogel and its good thermal insulation and heat preservation properties, its addition to the preparation of regenerated polyurethane can bring out the corresponding characteristics and make the prepared regenerated polyurethane foam have better performance. The thermal conductivity of the regenerated polyether polyol-based silica aerogel-regenerated polyurethane foam nanocomposites prepared with different silica aerogel additions is shown in Table 3.

As seen in Table 3, the thermal conductivity of each regenerated polyurethane foam nanocomposite meets the national standard of less than 0.036 W/(m·K). The addition of silica aerogel results in a lower thermal conductivity of the regenerated polyurethane foam nanocomposites compared to the regenerated polyurethane foam without the addition of silica aerogel. The overall decreasing trend of thermal conductivity with the increase in the amount of silica aerogel added indicates that silica aerogel has a significant optimization effect on the thermal conductivity of regenerated polyurethane foam nanocomposites, and the thermal insulation performance is enhanced to make better insulation materials. When the additional amount of silica aerogel reached 0.09 g, the thermal conductivity did not change much, and the thermal insulation effect was better. When the additional amount of silica aerogel was further increased to 0.12 g and 0.15 g, the thermal conductivity of the regenerated polyurethane foam increased and the strength began to decrease, which proved that the aerogel produced an agglomeration phenomenon and led to the decrease in thermal insulation performance; Therefore, the optimal addition of 0.09 g (0.4% of the total polyol) produced better silica-based nanocomposites of regenerated polyurethane foam with polyether polyol.

### 3.11. Thermal Weight Loss Test Analysis

TG analysis was performed using TGA-1000C (Shanghai Yingnuo Precision Instrument Co., Ltd., Shanghai, China). The thermal weight loss spectra of the regenerated polyurethane foam nanocomposites with different silica aerogel additions under a nitrogen environment are shown in Figure 13. It can be seen from the figure that the thermal weight loss of the regenerated polyurethane foam consists of three parts. The first part ranges from 0 to 200 °C, there is a slight decrease in the weight of the regenerated polyurethane foam at this stage, mainly due to water loss, and there is no significant difference at this point.

It can be seen from the figure that the second part ranges from 200 to 350 °C, the thermal weight loss of the regenerated polyurethane foam gradually decreases with the increase in the amount of silica aerogel added, and the temperature at which the regenerated polyurethane foam nanocomposites without and with 0.06 g of silica aerogel added start to decompose is about 225 °C, while the regenerated decomposition temperature of polyurethane foam with the addition of 0.09 g and 0.12 g of silica aerogel reached 275 °C. Thus, it can be seen that the addition of silica aerogel increases the thermal stability of the regenerated polyurethane foam nanocomposites. The third part is 350–500 °C. In this part, it can be seen that the residual rate of the regenerated polyurethane foam nanocomposites modified with the addition of 0.09 g and 0.12 g of silica aerogel has a significant increase and the rate of thermal weight loss has a significant decreasing trend compared with those without and with the addition of 0.06 g of silica aerogel foam, and with the increase in the amount of silica aerogel added, the regenerated thermal stability of the polyurethane foam nanocomposites was improved, and the residual rate would be close to 40% when the addition amount reached 0.12 g (0.4% of the total polyol).

Therefore, the higher the amount of silica aerogel added, the higher the residual rate of the regenerated polyurethane foam nanocomposites and the better their thermal stability.

### 3.12. Scanning Electron Microscopy Testing and Analysis of Silica Aerogel-Regenerated Polyol-Based Polyurethane Nanocomposites

The regenerated polyurethane foam/silica nanocomposite specimens prepared using 0.09 g of silica aerogel modification were sliced and compared with the pure samples of regenerated polyurethane foam prepared without the addition of silica aerogel and were observed and photographed together using scanning electron microscopy; the observed and photographed images are shown in Figure 14. From the figure, it can be seen that the pore structure of the regenerated polyurethane foam nanocomposites prepared by adding silica aerogel is more compact, the surface is smoother, the degree of closure is high, the structure is tougher, and the spacing between the pore walls is reduced, so that the gases can be better enclosed in order to achieve the effect of cold and heat insulation.

## 4. Conclusions

In this study, we successfully obtained liquid regenerated polyether polyol after the degradation of waste PU foam by the two-component decrosslinkers ethylene glycol and ethanolamine, and then successfully prepared silica aerogel and studied the effects of different ratios of silica aerogel additions on the various properties of regenerated polyether polyol-based polyurethane foam nanocomposites, and concluded as follows:The best ratio for the preparation of silica aerogel is when the mass ratio of ethyl orthosilicate: anhydrous ethanol: distilled water is 30:45:5, which is when the performance of silica aerogel is the best.Liquid regenerated polyether polyol was obtained by degradation of waste PU foam using the two-component decrosslinkers ethylene glycol and ethanolamine, and the re-generated polyol-based polyurethane foam was modified by adding different ratios of SiO_2_ aerogel to successfully prepare aerogel/regenerated polyurethane foam nanocomposites with SiO_2_ aerogel modified regenerated polyurethane. The regenerated polyurethane foam nanocomposites with a density of 34.1 kg/m^3^ and a compression strength of 0.301 MPa were prepared by changing the ratio to achieve the optimal reaction process conditions.The nanocomposites were modified by adding 0.09 g (0.3% of the total polyol) of silica aerogel; under optimal process conditions, the thermal conductivity was 0.0228 W/(m·K), the compressive strength was 0.585 MPa, and the density was 32.8 kg/cm^3^, at which time the comprehensive performance was better, and the bubble pores were more complete and close to a hexagonal shape, with thicker pore walls and a thicker skeleton as observed via scanning electron microscopy, and the best performance was achieved.

## Figures and Tables

**Figure 1 nanomaterials-13-02583-f001:**
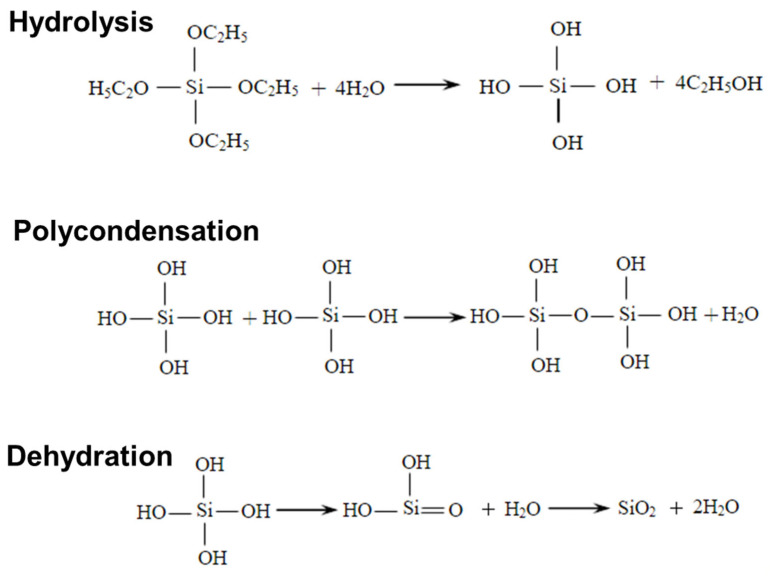
Preparation mechanism of silica aerogel.

**Figure 2 nanomaterials-13-02583-f002:**
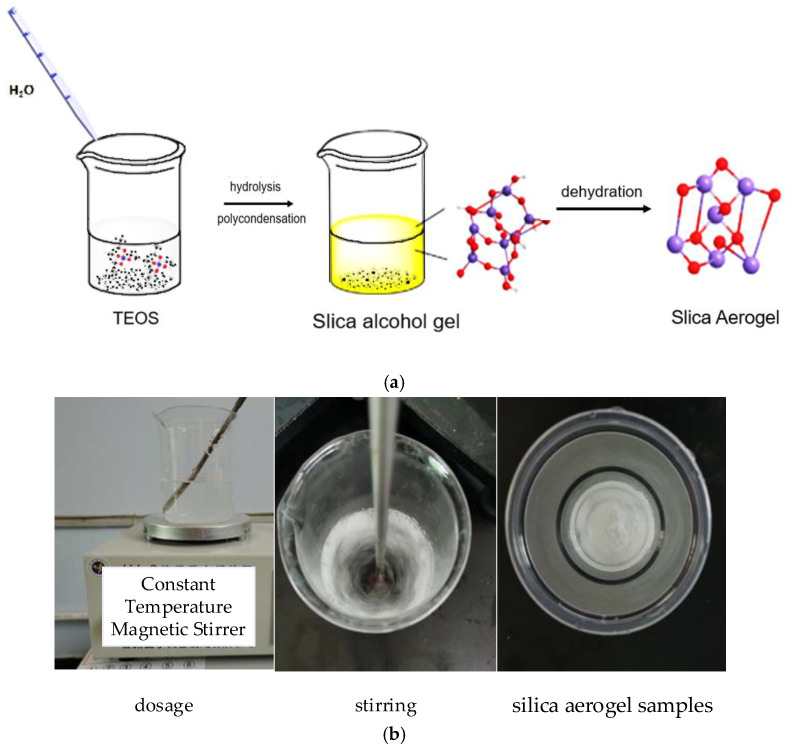
Silica aerogel preparation flow chart. (**a**) Schematic. (**b**) Actual photos of the process.

**Figure 3 nanomaterials-13-02583-f003:**
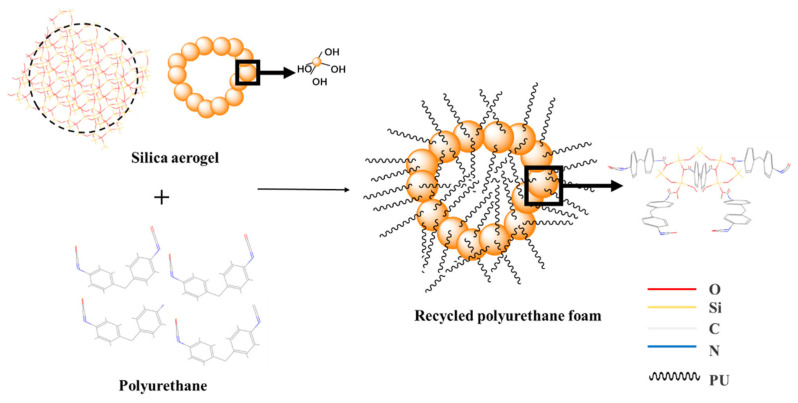
Schematic diagram of the reaction mechanism of hydroxyl-containing silica aerogels with isocyanates.

**Figure 4 nanomaterials-13-02583-f004:**
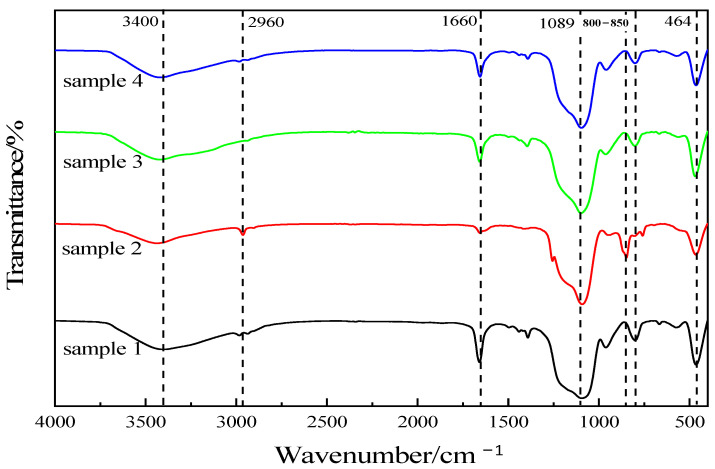
Infrared spectra of silica aerogels prepared with different drug ratios.

**Figure 5 nanomaterials-13-02583-f005:**
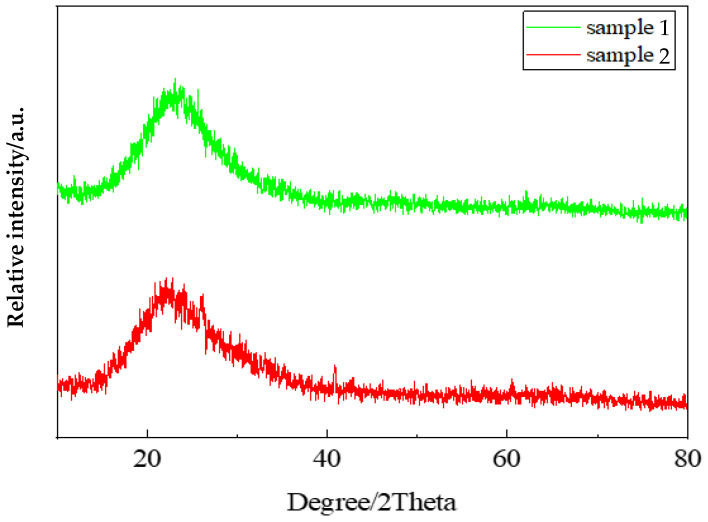
XRD pattern of silica aerogel. Sample 1: commercial sample; Sample 2: self-preparation sample.

**Figure 6 nanomaterials-13-02583-f006:**
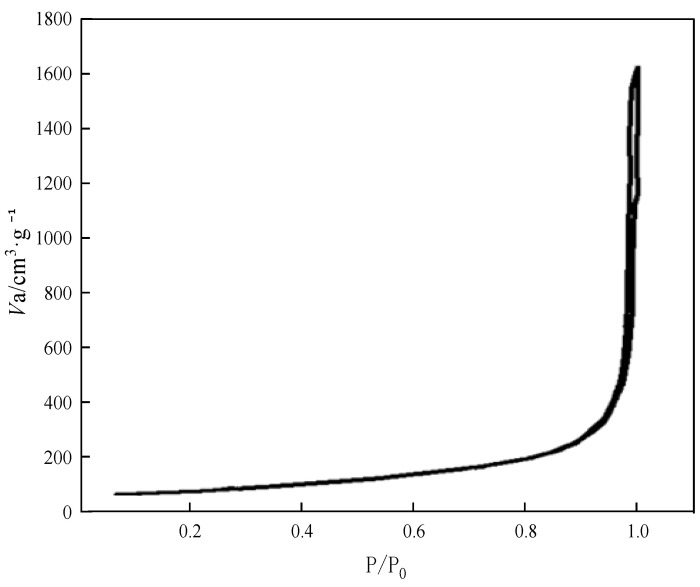
Absorption and desorption curves for aerogel samples.

**Figure 7 nanomaterials-13-02583-f007:**
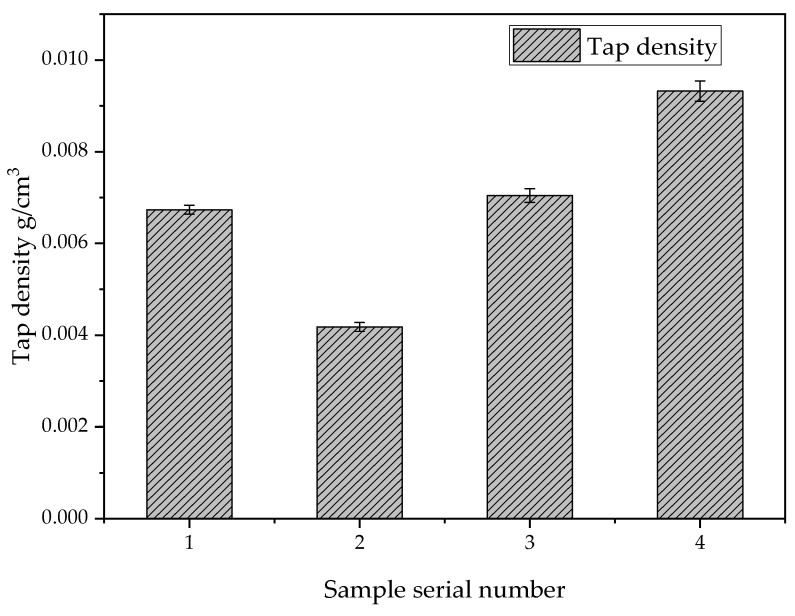
Tap densities of samples prepared with different drug ratios.

**Figure 8 nanomaterials-13-02583-f008:**
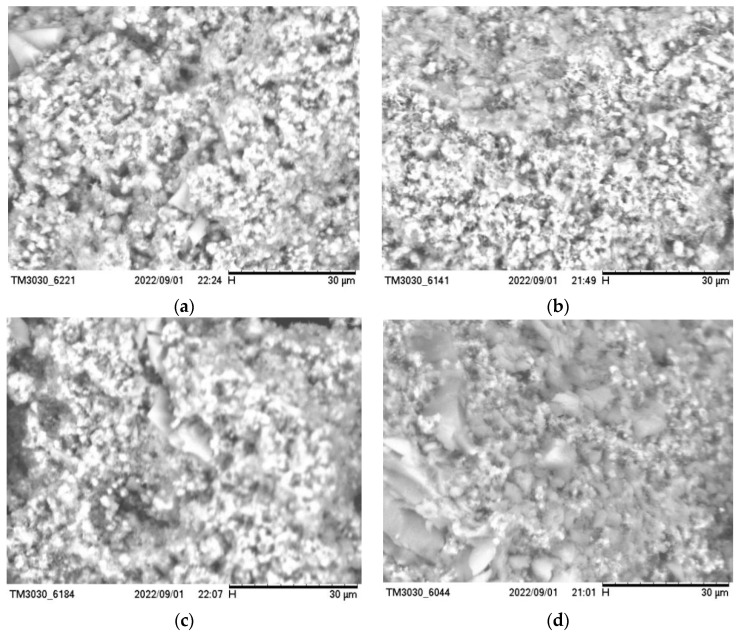
SEM images of different silica aerogels when (**a**) the mass ratio of ethyl orthosilicate, anhydrous ethanol, and distilled water was 15:45:5; (**b**) the mass ratio of ethyl orthosilicate, anhydrous ethanol, and distilled water was 30:45:5; (**c**) the mass ratio of ethyl orthosilicate, anhydrous ethanol, and distilled water was 45:45:5; (**d**) the mass ratio of ethyl orthosilicate, anhydrous ethanol, and distilled water was 60:45:5.

**Figure 9 nanomaterials-13-02583-f009:**
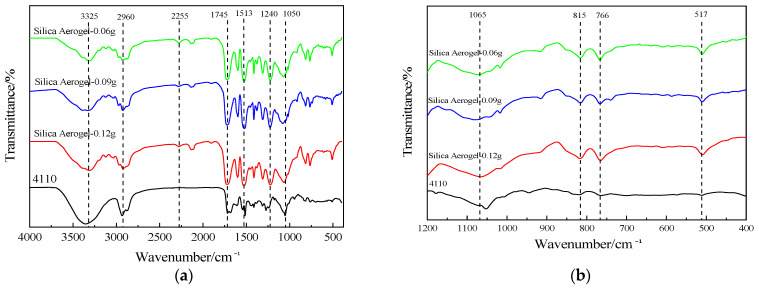
Infrared spectra of recycled polyurethane foam with different silica aerogel additions: (**a**) 4000–400 cm^−1^; (**b**) 1200–400 cm^−1^.

**Figure 10 nanomaterials-13-02583-f010:**
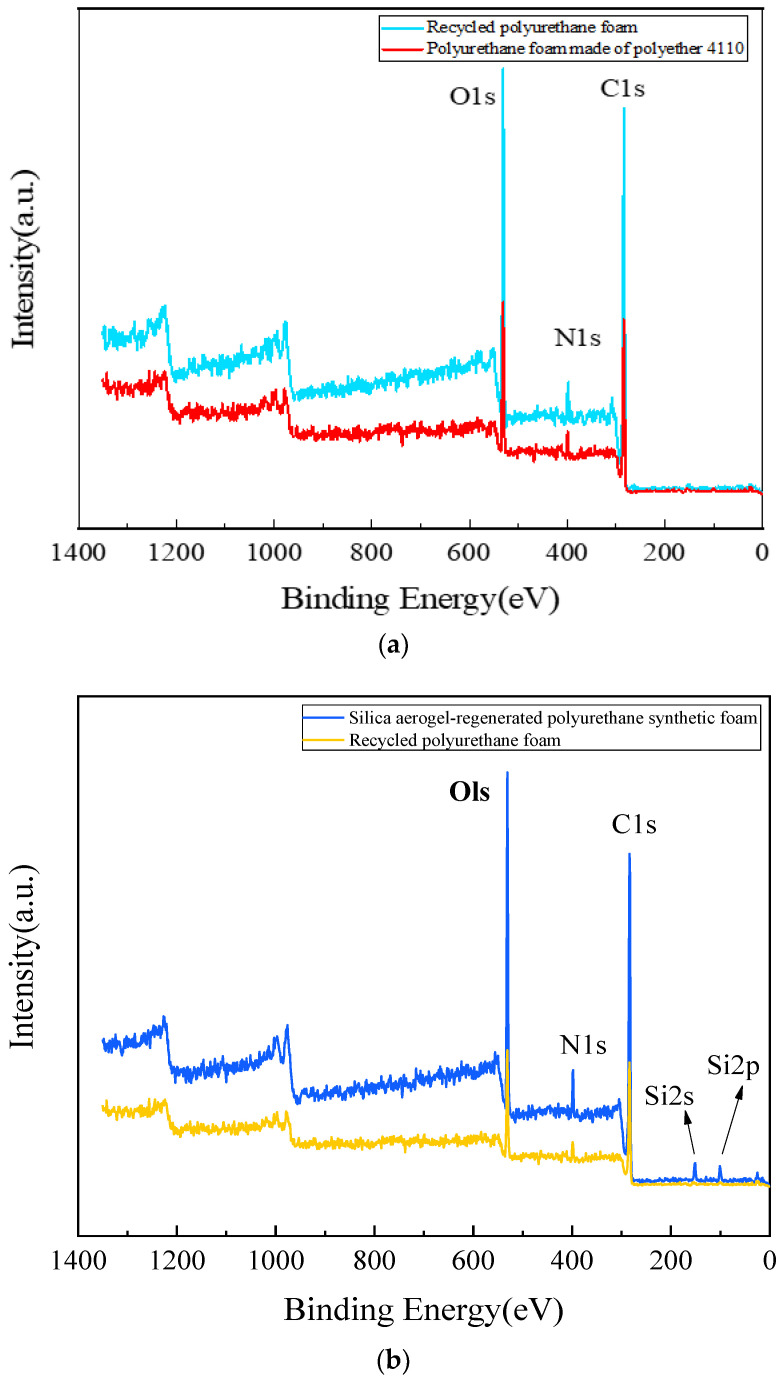
XPS spectrum of polyurethane foam. (**a**) the 4110 polyurethane foam and the regenerated polyurethane foam; (**b**)silica aerogel-regenerated polyurethane nanocomposite with 0.3% of the total mass of polyol and regenerated polyurethane foam without silica aerogel addition.

**Figure 11 nanomaterials-13-02583-f011:**
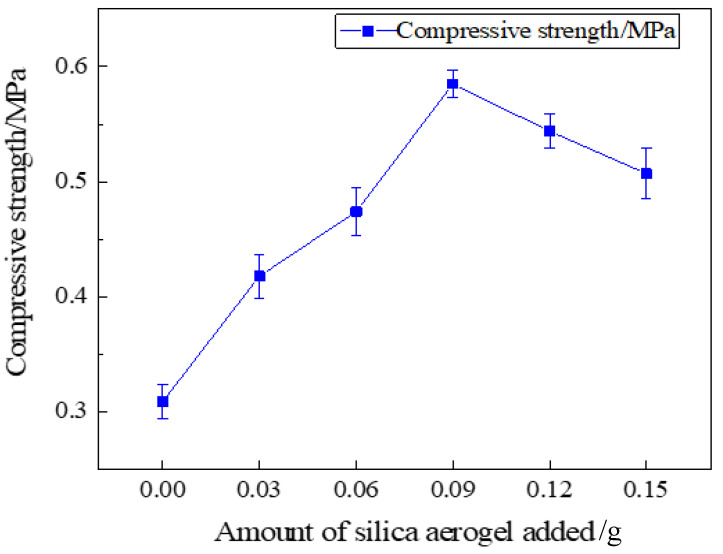
Compressive strength of recycled polyurethane foam with different aerogel additions.

**Figure 12 nanomaterials-13-02583-f012:**
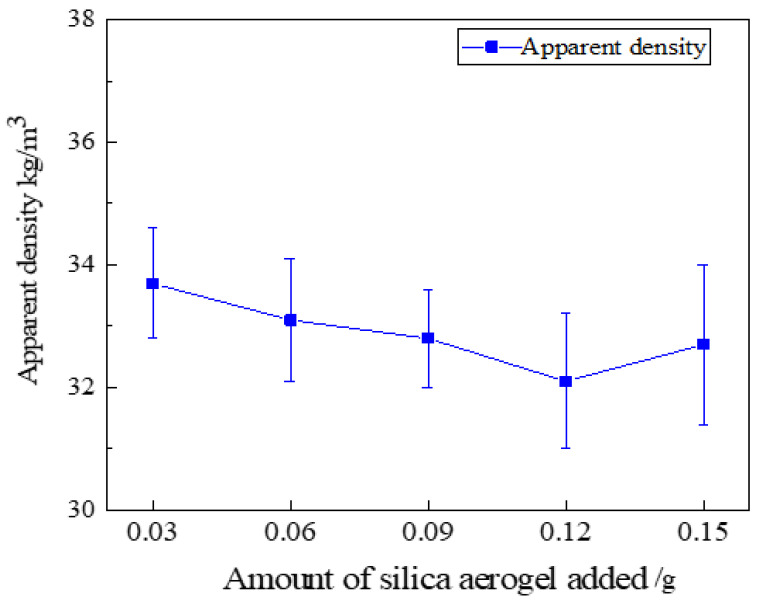
Apparent density of regenerated polyurethane foam with different silica aerogel additions.

**Figure 13 nanomaterials-13-02583-f013:**
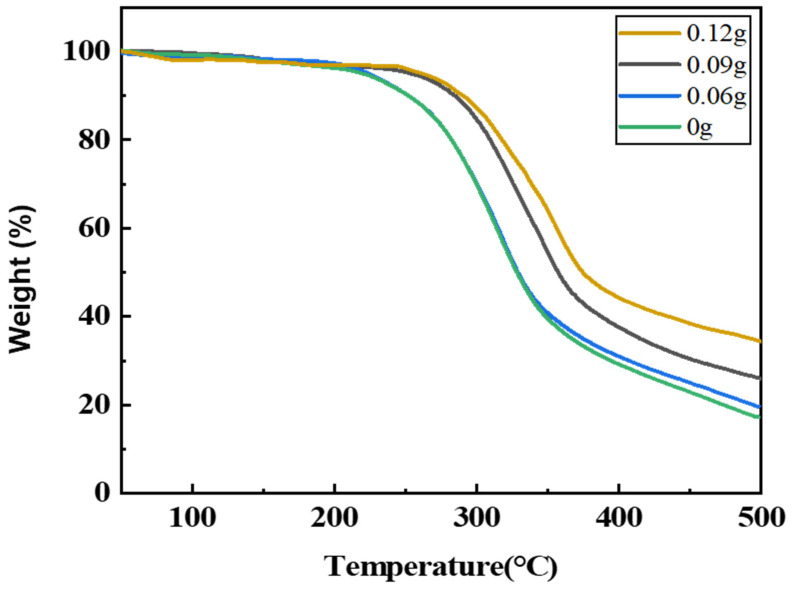
Thermogravimetric spectra of recycled polyurethane foams with different silica aerogel additions.

**Figure 14 nanomaterials-13-02583-f014:**
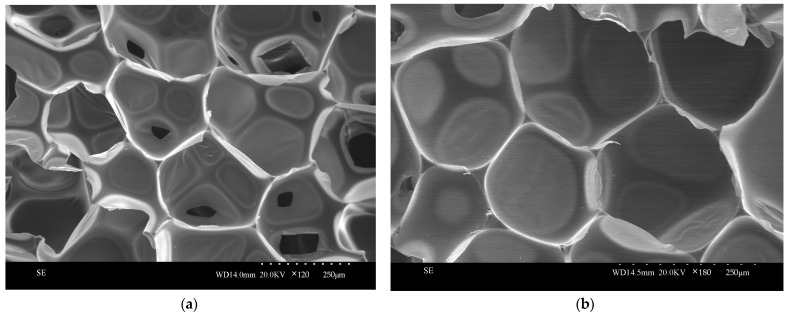
SEM images of recycled polyurethane foams prepared with different silica additions: (**a**) 0 g of silica aerogel; (**b**) 0.09 g of silica aerogel.

**Table 1 nanomaterials-13-02583-t001:** Dosage of medicines prepared by using silica aerogel.

Sample	Ethyl Orthosilicate (g)	Anhydrous Ethanol (g)	Distilled Water (g)
1	15	45	5.0
2	30	45	5.0
3	45	45	5.0
4	45	30	5.0

**Table 2 nanomaterials-13-02583-t002:** Silica aerogel-modified regenerated polyurethane foam formulation.

Reagents	Quantity (g)	Percentage (Total Mass of Polyol, %)
Recycled polyols	9.00	70.0
Polyether polyol 4110	21.00	30.0
Dimethylsilicone oil	0.60	2.0
Cyclopentane	9.00	30.0
Triethanolamine	0.30	1.5
Dibutyltin dilaurate	0.15	0.5
IPDI	39.00	130.0
Silicon dioxide aerogel	0.03, 0.06, 0.09, 0.12, 0.15	0.1, 0.2, 0.3, 0.4, 0.5

**Table 3 nanomaterials-13-02583-t003:** Thermal conductivity of recycled polyurethane foams prepared with different aerogel additions.

Amount of Silicon Dioxide Aerogel Added/g	Thermal Conductivity/W(m·K)^−1^
0	0.0297
0.03	0.0263
0.06	0.0244
0.09	0.0228
0.12	0.0229
0.15	0.0230

## Data Availability

The data presented in this study are available on request from the corresponding author.

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
