# Peer review of "Study of Aerogel-Modified Recycled Polyurethane Nanocomposites"

_nanomaterials, 2023, doi:10.3390/nano13182583_

Round 1

Reviewer 1 Report

The comments are embedded in the PDF file.

Reviewer 2 Report

The paper deals with the quite important aspect of polymer technologies – recycling of polymer materials. The authors developed SiO2 aerogel-modified regenerated polyurethane composites and performed series of experiments to characterize their properties. The results obtained are interesting and worth of being published in Nanomaterials. However it needs some corrections.

The overall impression is that the manuscript is written not very carefully.

In particular:

Section 2.3, page 5.

-        Please indicate the amount of silica aerogel added to the polyurethane foam, rather than write “certain amount”

-        Third paragraph. Change the description of the preparation procedure from a recipe-style (Take, place, mix…) to the commonly used style (The amount was taken, placed, mixed etc.). Indicate the stirring rate.

-        Table 2. Replace “quality” by “quantity”.

Section 3.1, page 7.

-        First paragraph. As shown in Figure 4, rather than 2.

Section 3.2, page 7.

“…both silica aerogel specimen 2(a) and silica aerogel specimen 3 (b) have diffuse diffraction peaks of low intensity and low peaks…”

-        (a) and (b) are not specified in Fig. 5.

-        What do you mean under “low peaks”?

-        The Y-axis in Fig. 5 does not have units at all

Section 3.3. First paragraph

-        - Again please change the style of procedure description.

-        - Remove part of a phrase at the end of this paragraph.

Section 3.4.

-        “microstructure of the silica aerogel at 1 μm…” what means at 1 μm?

-        Figure 7. Images are not indicated as (a), (b)… Please specify in the figure caption what samples are imaged and what is indicated by red circles.

Section 3.5.

-        Figs 8 and 9. It would be better to combine these into a single figure as (a) and (b) parts and mention clearly in the text that (b) is the enlarged part of (a).

-        Last phrase in Fig. 9. What do you mean under “nanocomposites prepared from polyether 4110”? If I understood correctly this sample does not contain aerogel.

 Section 3.7.

-        Figure 11. Please indicate units in the X axis.

Section 3.8

-        What is “cool place”? Please indicate the temperature.

-        Figure 12. Remove the word “different” from the legend in the X axis.

-        Correct the first phrase in the paragraph below Fig. 12. It is not understandable.

End of Section 3.9.

-        Again remove the unnecesary part of a phrase.

Section 3.11

-        Indicate samples corresponding to the images in the figure caption.  The discussion of the images is too concise, please expand.

Conclusions Section

-        Please indicate the concentrations of silica aerogel corresponding to the obtained best parameters of the composites.

Please check the language and make necessary corrections.

Round 2

Reviewer 1 Report

The authors of the paper have responded correctly to my questions and suggestions. Therefore, the submitted article is accepted

The English language of the work is appropiate